# Transcriptome Analysis of Deoxynivalenol (DON)-Induced Hepatic and Intestinal Toxicity in Zebrafish: Insights into Gene Expression and Potential Detoxification Pathways

**DOI:** 10.3390/toxins15100594

**Published:** 2023-10-02

**Authors:** Feng Yao, Miaomiao Zhao, Yaowen Du, Guoli Chang, Chuanpeng Li, Ruiyu Zhu, Chenggang Cai, Suqing Shao

**Affiliations:** 1College of Biological and Chemical Engineering, Zhejiang University of Science and Technology, Hangzhou 310023, China; 222103855017@zust.edu.cn (F.Y.); eliauk330@163.com (M.Z.); 222203855019@zust.edu.cn (Y.D.); changgl2022@163.com (G.C.); 222103855022@zust.edu.cn (C.L.); 115018@zust.edu.cn (R.Z.); 2Guelph Research and Development Centre, Agriculture and Agri-Food Canada, Guelph, ON N1G 5C9, Canada

**Keywords:** DON, zebrafish, liver toxicity, intestinal toxicity, transcriptome

## Abstract

The effects of deoxynivalenol (DON, 50 µg/mL) on the zebrafish liver and intestine were studied. Differentially expressed genes (DEGs) from mRNA and lncRNA were analyzed by RNA seq. Gene Ontology (GO) and signaling pathways were studied where the top 30 DEGs of each type of RNA were involved. The results showed there were 2325 up-regulated and 934 down-regulated DEGs of lncRNA in the intestinal tract, and 95 up-regulated genes and 211 down-regulated genes in the liver, respectively. GO functional annotation analysis showed that lncRNA was enriched in the biological processes, involving the RNA splicing, CSF1-CSF1R complexes, and MAP kinase activity. DEGs of lncRNA located in the KEGG signal pathways include the C-type lectin receptor signaling and the NOD-like receptor signaling pathways. Metabolism involves the biosynthesis of indole alkaloids, cancer pathways for human disease, MAPK and Rap1signaling pathways for environmental information processing, necroptosis and focal adhesion for cell processes. The mRNA gene expression analysis showed there were 1939 up-regulated, 1172 down-regulated genes and 866 up-regulated, 1211 down-regulated genes in the intestine and liver of zebrafish, respectively. This study provides transcriptome analysis and toxicological investigation of DON in the zebrafish liver and intestine, offering insights into gene expression patterns and potential detoxification pathways.

## 1. Introduction

Deoxynivalenol (DON) is a type B trichothecene toxin that was first purified and identified from moldy wheat by Japanese scientists Morooka and Yoshizawa [1] in 1972. DON is a secondary metabolite produced by *Fusarium graminearum* and *Fusarium flavum* when grains are infected with the Fusarium pathogens [2]. DON has a global impact on food contamination, and countries and regions such as China, Japan, South America, North America, and Canada are all high-incidence areas for DON, especially hot and humid areas near the equator [3]. Survey results from Biomin showed that the detection rates of DON pollution in East Asia, Nordic Europe, and Central America are the top three in the world, with 84.8%, 74.2%, and 70.0% detection rates, respectively. Among them, East Asia, including China, has a median average DON content of 418 μg/kg due to factors such as climate and geography, higher than the world average of 388 μg/kg [4]. The individual contamination of DON in China exceeded 96.4%, and the average concentration of DON in feed samples ranged from 458.0 to 1925.4 μg/kg [5]. Among those regulated mycotoxins, DON often contaminates grains (wheat, barley, oats, rye, and corn; less so rice, sorghum, and Triticale) and cereal food and feed. DON is one of the most widely distributed pollutants in human food and animal feed. In over 25,000 samples collected from 28 European countries between 2007 and 2014, DON was found in 47% of 4000 feed samples and in 45% of 1621 unprocessed grains with undetermined end-use [6]. Although DON is considered a non-carcinogenic compound [7], different countries have set threshold levels for this toxin in food and feed. For example, in piglet feed, the maximum limits for Europe, Canada, and the United States are 0.9, 1, and 5 mg/kg feed, respectively [8,9]. At the same time, DON is a serious hazard that not only reduces the yield and quality of crops but also has cytotoxicity, neurotoxicity, reproductive toxicity, carcinogenicity, and teratogenicity to animals [10]. People and animals suffer from diarrhea, vomiting, gastrointestinal inflammation, and other symptoms after ingestion of DON [11].

After mycotoxins are ingested through feed and food, the gastrointestinal tract is the first target organ. Liu et al. [12] found that the consumption of feed contaminated with ≥ 1.0 mg/kg DON caused piglet intestinal damage. Meanwhile, the DON-induced intestinal injury was a further impairment of redox homeostasis and ferroptosis signaling. WaśKiewicz et al. [13] found that small sows could accumulate DON in gastrointestinal tissue after taking low doses of DON orally over a short period of time. The concentration range of DON detected in small intestine samples was 7.2 ng/g (in the duodenum) to 18.6 ng/g (in the ileum) and 1.8 ng/g (in the transverse colon) to 23.0 ng/g (in the cecum) in large intestine samples, and the content range of DON in liver tissues was 6.7 to 8.8 ng/g. DON can affect the proliferation and vitality of animal and human intestinal epithelial cells. Ji [14] and other researchers found that the average daily weight gain and average daily feed intake of piglets fed DON-contaminated feed were significantly reduced, and the thickness of the smooth muscle layer in the jejunum and ileum and the expression of smooth muscle cell contraction markers were also reduced, which affected the contractility of smooth muscle cells and interfered with intestinal motility and growth performance. DON can also be adjusted by NF-κ. B and TOR pathways impair the intestinal immune function of grass carp juveniles [15]. Wang et al. [16] found that the mixture of AFB_1_ and DON elicited an additive combined effect on zebrafish embryos. The levels of CAT, caspase-3, and T4 markedly varied in most single and mixture groups. The expressions of four genes (cas3, apaf-1, cc-chem, and cyp19a) associated with oxidative stress, cellular apoptosis, the immune system, and the endocrine system were markedly varied upon the mixture exposure in comparison to the corresponding single exposure of AFB1 or DON.

Zebrafish (*Danio rerio*) has been widely used as a common model organism to evaluate the toxic effects of various mycotoxins on zebrafish embryos and adults due to various advantages, including easy observation and easy access to embryos [17]. Long term-feeding of 2.0 μg/kg DON can induce oxidative imbalance in zebrafish liver [18]. The liver is well known for its digestive and metabolic functions, and research has shown that it has abundant immune cells and is an important immune organ in living organisms [19]. After entering the animal body, fungal toxins act on the intestine, which has a certain tolerance to toxins and can alleviate the damage caused by fungal toxins to the body to a certain extent [20]. At present, there is no report on the toxicity of DON to adult zebrafish. This study uses adult zebrafish as an animal model for the first time to conduct intestinal and liver transcriptome analysis and toxicological study of DON, providing a theoretical basis for subsequent biological detoxification research.

## 2. Results

### 2.1. Differential Expression Analysis of lncRNA

A map of up-regulated differentially expressed genes of lncRNA and a volcano map of differentially expressed genes in zebrafish intestine and liver samples are shown in Figure 1. The number of up-regulated and down-regulated differentially expressed genes of lncRNA in the intestinal tract of zebrafish was 2325 and 934, respectively. In the liver of zebrafish, the number of up-regulated genes was 95, and the number of down-regulated genes was 211. The total number of differentially expressed genes of lncRNA in the intestine is 3259, and the total number in the liver is 306. It can be seen from the Venn diagram in Figure 2 that there are 124 differentially expressed genes of lncRNA shared by the intestine and liver of zebrafish. In addition to the co-expressed genes, 3135 differentially expressed genes of lncRNA can only be found in the intestine, and 182 differentially expressed genes of lncRNA can only be found in the liver. All the numbers are shown in Figure 1.

### 2.2. GO Functional Annotation Analysis of Differentially Expressed Genes

A GO functional annotation analysis of differentially expressed genes was conducted based on the GO database and included the classifications biological process (BP), cellular composition (CC), and molecular function (MF). It can be seen from Figure 3 and Figure 4 that, compared with the blank group, an intestinal difference comparison of the DON solution-treated zebrafish experimental group showed that lncRNA was enriched in the biological process classification and the set of differentially expressed genes included RNA splicing (73, Figure 3), definitive hemopoiesis (24), and β-positive regulation of beta amyloid formation (14); regarding the cell composition classification, the set of differentially expressed genes mainly included CSF1-CSF1R complexes (6); and for the molecular function classification, the set of differentially expressed genes mainly included MAP kinase activity (15) rho-dependent protein serine/Threonine kinase activity (10), JUN kinase activity (8), etc. The number of differentially expressed genes is listed in Figure 3.

Comparisons of liver differences between the experimental group and the blank group regarding lncRNA enrichment in the biological process classification of differentially expressed genes mainly include NAD biosynthetic processes (8), cell development (7), myelocyte differentiation (7), etc. The classification of cell composition of differentially expressed genes mainly includes core-binding factor complexes (7), CSF1-CSF1R complexes (5), and BRISC complexes (4). The classification of the molecular function of differentially expressed genes mainly includes monooxygenase activity (8), cytokine binding (5), NADPH: sulfur Oxidoreductase activity (5), etc. The number of differentially expressed genes is listed in Figure 3.

### 2.3. KEGG Signal Pathways Analysis

The KEGG signal pathways of the top 30 differentially expressed genes of the lncRNA of zebrafish intestinal and liver transcriptomes are shown in Figure 5 and Figure 6. The intestinal samples include five categories: biological system, metabolism, human disease, environmental information processing, and cell processes. On this basis, the liver samples have one more category of genetic information processing.

Compared with the control group and the blank group, the KEGG signal pathways where in differentially expressed genes of lncRNA are located include the C−type lectin receptor signaling pathway, the NOD−like receptor signaling pathway, the chemokine signaling pathway, and the platelet activation and relax in signaling pathway. Metabolism involves the biosynthesis of indole alkaloids. Human diseases mainly include cancer pathways, i.e., influenza A, pertussis, Proteoglycans in cancer, and hepatitis C. Environmental information processing mainly includes the MAPK signaling pathway, the Rap1 signaling pathway, and the ErbB signaling pathway. Cell processes mainly include necroptosis, focal adhesion, and regulation of the actin cytoskeleton.

Compared with the blank group, the KEGG signaling pathways of the differentially expressed genes of lncRNA in the liver-sample control group include the NOD-like receptor signaling pathway, bile secretion, ovarian steroidogenesis, etc., in the biological system classification. Metabolism mainly includes nicotinate and nicotinamide metabolism, fatty acid elongation, tyrosine metabolism, and biosynthesis of unsaturated fatty acids. Human diseases mainly include Cushing’s syndrome, acute myeloid leukemia, toxoplasmosis, pancreatic cancer, etc. Genetic information processing mainly involves SNARE interactions in vesicular transport. Environmental information processing mainly includes cytokine–cytokine receptor interaction, the HIF−1 signaling pathway, and the TGF beta signaling pathway. Cellular processes mainly involve signaling pathways regulating the pluripotency of stem cells and adhesive junctions.

### 2.4. mRNA Gene Expression Analysis

A map of up-regulated mRNAs of differentially expressed genes and a volcano map of differentially expressed genes in zebrafish intestine and liver samples are shown in Figure 7. The number of mRNAs of differentially expressed genes in the intestine of zebrafish was 1939 (Figure 7, the same as below of this section), and the number of down-regulated genes was 1172. In zebrafish liver, the number of up-regulated genes was 866, and the number of down-regulated genes was 1211. The total number of mRNAs of differentially expressed genes in the intestine is 3111, and the total number in the liver is 2077. It can be seen from the Venn diagram in Figure 8 that there are 480 mRNAs of differentially expressed genes shared by the intestine and liver of zebrafish. In addition to the co-expressed genes, 2631 mRNAs of differentially expressed genes can only be found in the intestine, and 1597 mRNAs of differentially expressed genes can only be found in the liver.

### 2.5. GO Enrichment Analysis of Differential Comparison Gene mRNA

A GO enrichment analysis of differential comparison gene mRNA was carried out. It can be seen from Figure 9 and Figure 10 that, compared with the blank group, the intestinal differential comparison gene mRNAs of the experimental group of zebrafish treated with DON were enriched in the biological process classification and the set of differential genes mainly included negative regulation of transcription from RNA polymerase II promoter (98), cell differentiation (87), positive regulation of transcription, DNA templated (81), etc. The set of gene mRNAs in the cell composition classification and comparison mainly included those of the nucleus (732), cytoplasm (685), cytosol (549), nucleoplasm (278), extracellular space (136), etc. In the molecular function classification and differential comparison, the set of gene mRNAs mainly included ATP binding (281), sequence-specific DNA-binding transcription factor activity (105), sequence-specific DNA binding (84), receptor binding (48), iron ion binding (31), etc. The number of differentially expressed genes is listed in Figure 9.

Regarding the comparison of liver differences between the experimental group and the blank group, mRNA enrichment in the biological process classification and the set of differentially expressed genes mainly include response to estradiol (31), response to insulin (29), cellular response to heat (26), response to polycyclic arene (24), determination of left/right symmetry (21), etc. The set of gene mRNAs in the cell composition classification and comparison mainly includes cytoplasm (465), axoneme (24), yolk (24), motile cilia (15), dynein complex (14), etc. The set of mRNAs of the molecular function classification and difference comparison genes mainly include receptor binding (36), lipid transporter activity (26), nutrient reservoir activity (26), dynein light chain binding (16), dynein intermediate chain binding (14), etc.

### 2.6. KEGG Signal Pathways Analysis

The KEGG signal pathways of the top 30 differential gene mRNA rankings of zebrafish intestine and liver transcriptomes are shown in Figure 11 and Figure 12. The intestinal samples include six categories: biological system, metabolism, human disease, genetic information processing, environmental information processing, and cell processes. On this basis, the liver samples lack a major category of genetic information processing.

Compared with the control group and the blank group, the KEGG signal pathways of the differential gene mRNAs in the intestinal sample include the thyroid hormones signaling pathway, complement and coagulation cascades, the PPAR signaling pathway, etc., in the biological system category. The metabolism category includes lysine degradation, cysteine and methionine metabolism, the Pentose phosphate pathway, etc. Human diseases mainly include microRNAs in cancer, insulin resistance, the AGE−RAGE signaling pathway in diabetic complications, etc. Genetic information processing mainly includes ribosomes, proteasomes, etc. Environmental information processing mainly includes the FoxO signaling pathway, the HIF−1 signaling pathway, the Hippo signaling pathway of multiple species, etc. The main cellular processes include the p53 signaling pathway and ferroptosis, etc.

Compared with the control group and the blank group of liver samples, the KEGG signal pathways of differential gene mRNAs include the NOD−like receptor signaling pathway, the C−type lectin receptor signaling pathway, the insulin signaling pathway, etc., in the biological system classification. Metabolism mainly includes lysine degradation, arginine and proline metabolism, starch and sucrose metabolism, etc. Human diseases mainly include pertussis and insulin resistance, etc. Environmental information processing mainly includes the FoxO signaling pathway, the AMPK signaling pathway, two-component systems, etc. The main cellular processes include necroptosis and biofilm formation, *Escherichia coli*, etc.

## 3. Discussion

Recent research results indicated that over 70% of the world’s grains are contaminated with fungal toxins [21,22], typically mixtures [23]. Koletsi et al. [24] summarized the existing knowledge on the effects of DON on farmed fish species and evaluated the risk of DON exposure in fish, based on data from in vivo studies. Consumption of DON-contaminated feeds by fish, even at levels below the European Commission (EC) recommendation limit (5000 µg/kg), can result in adverse although non-lethal effects on fish such as impaired feed intake, growth performance, immunity, detoxification capacity, and tissue damage and oxidative stress. Exposure to high concentrations of DON is associated with diarrhea, vomiting, increased white blood cells, and gastrointestinal bleeding. Long-term exposure can affect the growth, immunity, and intestinal barrier function of animals [25,26,27]. The toxin interacts with the Peptidyl transferase region of the 60S ribosome subunit to induce “ribotoxic stress”, resulting in the activation of mitogen-activated protein kinase (MAPK) and its downstream pathways [25,28].

In this study, zebrafish was used as a model to study the effects of environmental DON exposure on the liver and intestine. Transcriptome sequencing technology was used to compare gene expression difference between a blank control group and a DON experimental group (50 µg/mL). In a comparison of lncRNA, the differentially expressed genes before and after the treatment of DON changed significantly, and DON had a greater impact on the liver and intestine. Based on the GO database, GO function annotation analysis of differentially expressed genes was carried out. Compared with the blank group, the intestinal differential comparison of the experimental group of Zebrafish treated with a DON solution was enriched in the biological process classification and the set of differentially expressed genes mainly included RNA splicing (73) and deterministic hematopoiesis (24). This may be because DON inhibits RNA splicing and intestinal deterministic hematopoietic functions. Yuan [29] et al. showed that DON induces pre-mRNA alternative splicing in HepG2 cells and inhibits the expression level of splicing factors U2AF1 and SF1. Analysis of the liver transcriptome of zebrafish shows that DON treatment triggers liver stress reactions and changes the expression of multiple genes related to NAD biosynthesis, cell development, and myelocyte differentiation. DON has strong cytotoxicity, mainly acting on rapidly growing and dividing cells, so cell development and differentiation are greatly affected. Ren [30] et al. found that DON can induce the expression of oxidative stress indicators ROS and malondialdehyde (MDA) in pig spleen lymphocytes, thereby inhibiting the antioxidant capacity of cells. KEGG signal pathway analysis of the top 30 differential genes of lncRNA in the intestinal and liver transcriptomes of zebrafish found that the cancer pathway gene after DON treatment changed the most. After DON entered the body, its active metabolite could alkylate with genetic material of the body, leading to damage to the genetic material and abnormal expression of tumor suppressor genes. Huang [31] and others found that DON has certain carcinogenic effects on mice.

The total number of mRNAs of differentially expressed genes in the intestinal tract of zebrafish was 3111; the total number in the liver was 2077. The number of mRNAs of up-regulated differentially expressed genes in the intestinal tract of zebrafish was 1939, and the number of down-regulated genes was 1172. In zebrafish liver, the number of up-regulated genes was 866 and the number of down-regulated genes was 1211. GO enrichment analysis of differentially expressed gene mRNA showed that the greatest difference in gene expression was found in the cell composition of the intestine, indicating that DON can affect cell composition in the intestine. Akbari et al. [32] confirmed that the concentration is as low as 1.39 μM. The monolayer of human Caco-2 can be disintegrated when exposed to DON for less than 1 h. Therefore, some researchers believe that DON toxins mainly reduce ion transfer between intestinal cells and disrupt the integrity of intestinal cells by altering the intestinal TEER (transmembrane epithelial resistance) [33]. Springler et al. [34] found through further research that the effect of DON on TEER is at least partially mediated by p44/42, and the epithelial cell barrier may be one of the susceptibility factors leading to inflammatory diseases. Compared with the blank group, the liver differential comparison mRNA of the experimental group is enriched in the biological process classification and the set of differential genes include major powerful protein complexes, lipid transport protein activity, dynein light chain binding, and dynein intermediate chain binding. ZO-1 (zona occludens1) protein and claudin-4 protein are two important components of the tight node structure, which play an important role in forming and maintaining the normal function of the intestinal barrier [28]. Wang et al. [35] studied weaned piglets and found that the distribution and mean optical density of ZO-1 protein in the intestinal tissue of the DON-treated group decreased. De Walle et al. [36] found that DON has dual toxicological effects on differentiated Caco-2 (human colon adenocarcinoma cells), including inhibiting protein synthesis and increasing monolayer cell permeability, and this toxic effect is likely to be achieved by reducing the synthesis of claudin-4 protein, an important component of TJ. In the KEGG signal pathways of the top 30 differentially expressed genes in zebrafish intestine and liver transcriptome mRNA, DON can affect the immune pathway in the liver intestine, leading to an immunosuppressive effect, and DON can affect the thyroid gland and some other pathways, leading to the emergence of inflammation. Mao et al. [37] found that DON can induce the activation of the Caspase-3/GSDME pathway to mediate cell apoptosis and induce the occurrence of liver inflammation in mice. DON also affects the metabolism of zebrafish, which can change mRNA expression, protein (enzyme) levels, and cytokines (IL-6, IL-10, and TNF- α). The number of apoptosis-related genes (Caspase-3) reduces the synthesis of hematopoietic cell kinase (HCK).

## 4. Conclusions

This study represents the first investigation into the toxicological effects of DON on adult zebrafish. By focusing on the intestine and liver of zebrafish and conducting GO enrichment and KEGG pathway analyses of metabolomes and transcriptomes, we gained valuable insights into the impact of DON on the metabolic pathway in zebrafish intestines and the gene signaling pathways in both the intestine and the liver. Our findings offer a preliminary understanding of the adverse reactions caused by DON in organisms.

The detoxification of DON presents formidable challenges and complexities. However, the insights gained from this toxicological assessment of zebrafish can serve as a significant step for the establishment of zebrafish models in future studies, particularly in investigating detoxification mechanisms within the intestinal tract and liver. This work contributes to advancing our knowledge of the toxicological effects of DON and opens avenues for future research aimed at mitigating its harmful consequences on organisms.

## 5. Materials and Methods

### 5.1. Chemicals and Reagents

Deoxynivalenol (DON (≥ 99.9%)) was purchased from Triple Chemical Corp. Ltd., Guelph, ON, Canada. Zebrafish were purchased from Shanghai FishBio Corp. Ltd., Shanghai, China. CaCl_2_, MgSO_4_, NaHCO_3_, and KCl (AR) were purchased from Maclean’s Biochemical Technology Co., Ltd., Shanghai, China. DEPC and Diethyl pyrocarbonate was purchased from Sinopharm Chemical Reagent Co., Ltd., Shanghai, China. TRIzol was purchased from Invitrogen Co., LTD. Chloroform, ethanol, isopropyl alcohol and isoamyl alcohol were purchased from Xilong Chemical Co., Ltd., Shanghai, China.

### 5.2. Pretreatment of Zebrafish

Preparation of standard dilution water: take 25 mL of 11.76 g/L CaCl_2_ solution, 4.93 g/L MgSO_4_ solution, 2.50 g/L NaHCO_3_ solution, and 0.23 g/L KCl solution, mix them, use ultrapure water to fix the volume to 1 L, and adjust the pH to 7.8 ± 0.2.

The average weight of the zebrafish tested was about 0.3 ± 0.1 g, and the body length was about 3. 5 ± 0.3 cm. The zebrafish were fed twice a day for one week in standard diluted water at 20 to 25 °C. Stop feeding 24 h before the experiment, and do not feed during the whole toxic exposure period. The standard dilution water was used to prepare a DON solution with a concentration of 50 µg/mL to feed the zebrafish for 96 h and set a blank group at the same time. The feeding density of zebrafish was 60 pieces/L DON solution (the blank group was standard dilution water). After feeding, the zebrafish was placed in in ice water for euthanasia, the intestines and livers were dissected quickly on ice, after cleaned them with DEPC water (ultrapure water treated with diethyl pyrocarbonate and sterilized at high temperature and high pressure), the samples were placed in a 3 mL cryopreservation tube, then placed the dissected intestines in liquid nitrogen quickly, and transferred them to a −80 °C for preservation. Every 60 zebrafish livers were divided into 1 group for liver transcriptome analysis, and 3 replicates were set up in the treatment group. Dry ice was used to transport the samples to Suzhou GENEWIZ Biotechnology Co., Ltd., Suzhou, China. All rearing and treatment were carried out in strict accordance with rules by the Institutional Animal Care and Use Committee (IACUC) of Hangzhou Normal University (Approval No. HSD 20221204) on 4 December 2022 (Hangzhou, China).

### 5.3. Extraction and Detection of Total RNA from Samples

The transcriptome sequencing process includes RNA extraction, RNA sample quality detection, library construction, library purification, library detection, library quantification, generation of sequencing clusters, and computer sequencing. The extraction and purification of total RNA from zebrafish larva samples were completed by TRIzol reagent method, and the operation steps were carried out according to the kit instructions. The detailed operations were as follows: (1) Fragmentation of tissue samples, namely, 1.5 mL of TRIzol lysate was added into 2 mL grinding and crushing tube containing zebrafish liver or intestinal samples; the Trizol lysate was ground in TissueLyser II grinder (QIAGEN China (Shanghai) Co., Ltd., Shanghai, China) for 30 s and left at room temperature for 5 min until the samples were fully cracked; (2) Sample extraction and purification: the fully cracked sample was centrifuged at 4 °C for 5 min at 12,000× *g*, then the supernatant was transferred into a new 1.5 mL RNase-free centrifuge tube, after addition of 300 μL chloroform/isoamyl alcohol (24:1, *v*/*v*) and mixed upside down, the cracking mixture was centrifuged at 4 °C and 12,000× *g* for 8 min, then the supernatant was transferred into a new 1.5 mL RNase-free centrifuge tube; then 600 μL of isopropyl alcohol was added and mixed thoroughly, the tube was placed at −20 °C for 2 h, after resting at low temperature, the sample was centrifuged at 4 °C and 17,500× *g* for 25 min, the supernatant was discarded after centrifugation, the precipitate was washed with 900 μL pre-cooled 75% ethanol and the sample was centrifuged at 4 °C and 17,500× *g* for 3 min, the supernatant was discarded and tube was dried on a super clean workbench (about 5 min), then the sample was dissolved and precipitated with 100μL DEPC treated water; (3) Total RNA concentration and quality detection: the concentration of total RNA (the average concentration of all extracted RNA samples was 609.47 ± 116.82 ng/μL) and the ratio of 28S/18S (the average ratio of samples was 1.93± 0.11) were measured by Agilent 2100 biochip analyzer (Agilent Technology Co., Ltd., Palo Alto, CA, USA) and RNA Integrity Number (RIN; The average RIN value of the samples was 9.96 ± 0.05, all of which were qualified samples and could be used for subsequent computer sequencing and analysis).

### 5.4. Construction of cDNA Library

Total RNA samples were processed based on the rRNA removal method, DNA probes were used to hybridize rRNA, and DNA/RNA hybridization chains were selectively digested by RNaseH. Then, the DNA probe was digested by DNase I, and the required qualified RNA sample was obtained after purification. Fragment the RNA sample obtained in last step using the interrupt buffer; then, a reverse transcription reaction was performed with a random N6 primer to synthesize double-stranded DNA; the double-stranded DNA ends were patched and the 5 ‘end was phosphorylated, forming an “A” sticky end of the 3’ end and connecting a 3 “T” connector; The linked products were amplified by PCR through specific primers; after thermal denaturation of the PCR amplification product into a single strand, it was cyclized by a bridge primer, that is, a single strand circular DNA library was obtained. Then, the computer was sequenced and analyzed.

### 5.5. High-Throughput RNA-Seq Sequencing and Screening of Differentially Expressed Genes (DEGs)

In this study, high-throughput transcriptome RNA-Seq sequencing was performed by Illumina Hiseq (Suzhou GENEWIZ Biotechnology Co., LTD., Suzhou, China). To ensure the accuracy and repeatability of RNA-Seq sequencing results, the machine samples of each treatment group were provided with 3 biological replicates, each containing 60 intestinal or liver samples, and the sequencing and results were analyzed by Genewiz Corporation (Suzhou GENEWIZ Biotechnology Co., LTD., Suzhou, China) using Illumina technology.

Raw data obtained by GWZHISEQ01 sequencing (Suzhou GENEWIZ Biotechnology Co., LTD., Suzhou, China) are called Raw Reads. The average output of raw data of all samples in this paper was 24.13 M. The original sequencing sequence was filtered through the filtering software SOAPnuke (v1.5.2, https://github.com/BGI-flexlab/SOAPnuke) (accessed on 27 November 2020) to remove the sequences containing joints, unknown base N content greater than 10% and low quality, etc, then the filtered sequencing sequences (Clean Reads) were obtained. The Hierarchical Indexing for Spliced transcripts was performed using the HISAT2: The Alignment of Transcripts method [38] aligns the filtered sequencing sequence to the zebrafish reference genome sequence (NCBI GRCz11, NCBI GRCZ11, NCBI transcripts). (https://www.ncbi.nlm.nih.gov/assembly/GCF_000002035.6 (accessed on 9 May 2017); then, the filtered sequencing sequence was compared to the zebrafish reference coding gene set by Bowtie 2 (v 2.2.5) method [39]. Based on the FPKM (the Fragments Per Kilobase per Million mapped fragments) method as an indicator to measure gene expression, the expression abundance of sequenced transcripts was calculated and quantified by RSEM (RNA-Seq by Expectation-Maximization) software package [40]. The DESeq 2 (v 1.4.5) R language package [41] was used to identify and analyze the differentially expressed genes (DEGs) among each treatment group, and the Fold Change that met the abundance of gene expression was screened through further calculation and analysis. DEGs with twice or more times adjusted *p*-value (Padj) less than or equal to 0.05 were used for subsequent gene function annotation and enrichment analysis.

### 5.6. Gene Ontology (GO) and KEGG Functional Enrichment Analysis

By using the DAVID biological information resources online analytical tools (v 6.8) [42] in 5.5, the DEGs of screening and Gene Ontology (Gene Ontology, GO) database (http://www.geneontology.org (accessed on 1 January 2019) for comparative analysis, the biological functions involved in these DEGs were annotated by GO classification, including three categories: biological process, cellular component and molecular function. After zero-mean normalization of the number of DEGs for each GO term, the bubble map is drawn by combining the number of z-scores, Padj, and DEGs for further visual analysis of the results. Meanwhile, Metascape [43] was used to compare the screened DEGs with the KEGG public database (https://www.kegg.jp/, accessed on 1 January 1995) [44] and was compared to analyze the enrichment of the signaling pathways or metabolic pathways involved in these DEGs.

### 5.7. Differential Expression Analysis of lncRNAs and mRNAs

The FPKMs of all transcripts in each sample were calculated by using StringTie (version 1.3.1). The differential expression analysis of lncRNAs and mRNAs between two conditions was performed using the DESeq2 (version 1.10.1) package in the R project. The transcripts with *p* < 0.05 and absolute FC > 2.0 were considered DE.

### 5.8. Data Processing and Analysis

The GO enrichment analysis results screened by DAVID were plotted into bubble maps using OriginPro 2019b (OriginLab, Northampton, MA, USA) software to more intuitively show the GO terms that were significantly enriched after exposure treatment. Metascape was used to conduct KEGG enrichment analysis for up-regulated and down-regulated DEGs, and the top 30 KEGG pathways for up-regulated and down-regulated DEGs enrichment were screened out. The enrichment results were mapped using the R language ggplot2 software package [45].

## Figures and Tables

**Figure 1 toxins-15-00594-f001:**
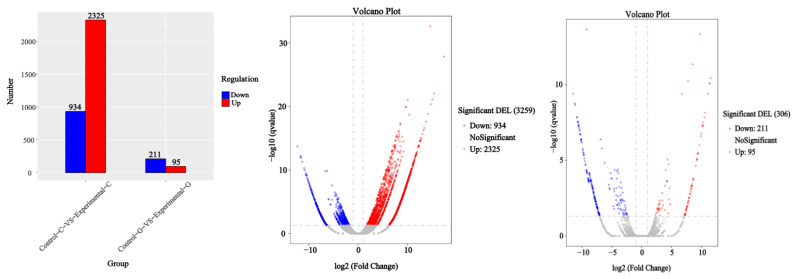
Comparison of sample differences in up-regulated differential expression genes of lncRNA and volcano map.

**Figure 2 toxins-15-00594-f002:**
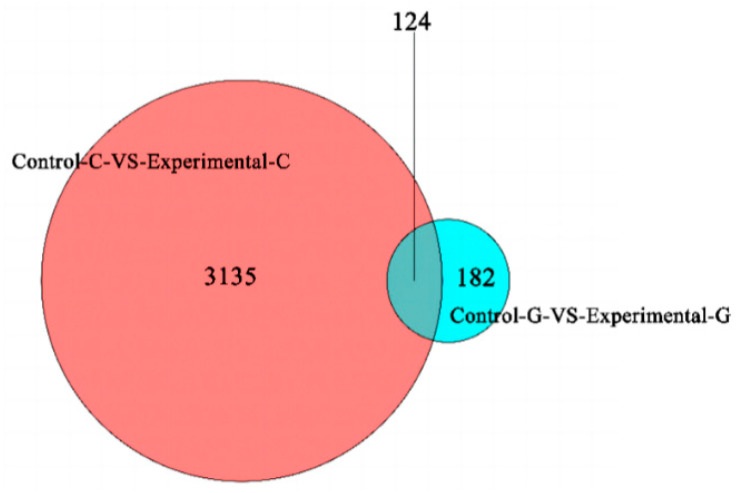
Venn diagram of differential expression genes of lncRNA.

**Figure 3 toxins-15-00594-f003:**
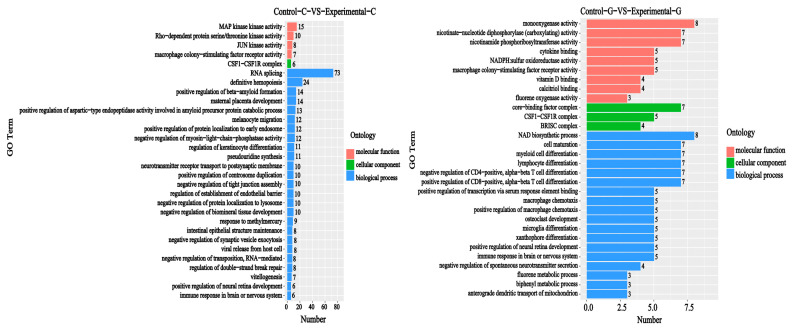
Sample difference comparison: lncRNA GO enrichment histogram. The vertical axis represents the enriched GO term, and the horizontal axis represents the number of differentially expressed genes in the term. Different colors are used to distinguish biological processes, cellular components, and molecular functions. In the Figure 3, mitogen-activated protein (MAP) kinase kinase activity, Jun kinase (JUN) activity, Colony-stimulating factor 1-Colony-stimulating factor 1receptor (CSF1-CSF1R) complexes, nicotinamide adenine dinucleotide phosphate hydrogen (NADPH), BRCC36 interacting with SCC1 (BRISC) complexes, nicotinamide adenine dinucleotide (NAD) biosynthetic processes.

**Figure 4 toxins-15-00594-f004:**
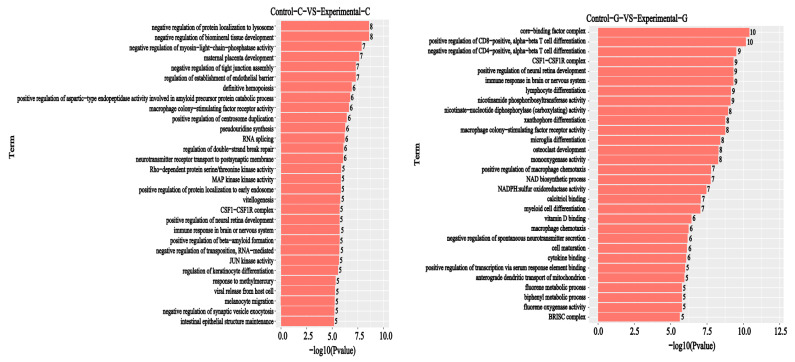
Sample difference comparison: lncRNA GO enrichment *p* value histogram, with the vertical axis representing the enriched GO term and the horizontal axis representing the log10 (*p*−value) value.

**Figure 5 toxins-15-00594-f005:**
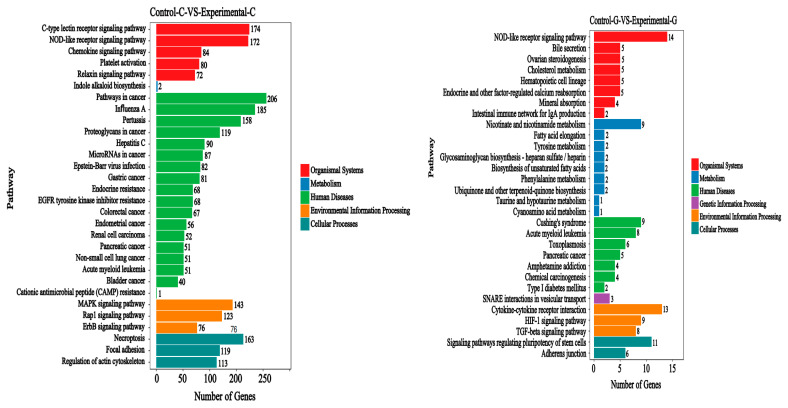
Sample difference comparison. Top 30 KEGG pathways: annotation classification Bar chart. The vertical axis represents the path name, and the horizontal axis represents the number of genes. The mitogen-activated protein kinase (MAPK) signaling pathway, Ras-related protein 1(Rap1) signaling pathway, erythroblastic oncogene B (ErbB) signaling pathway, nucleotide oligomerization domain-like (NOD−like) receptor signaling pathway, hypoxia inducible factor−1 (HIF−1) signaling pathway.

**Figure 6 toxins-15-00594-f006:**
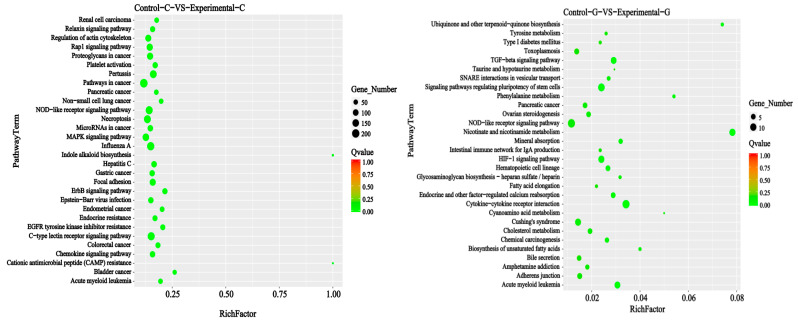
Scatter plot of lncRNA KEGG enrichment of differentially expressed genes. The vertical axis represents the path name, the horizontal axis represents the Rich factor, and the size of the dot indicates the number of differentially expressed genes in this path. The color of the dot corresponds to different Q value ranges. Soluble N-ethylmaleimide-sensitive factor attachment protein receptor (SNARE) interactions in vesicular transport, hypoxia inducible factor−1 (HIF−1) signaling pathway, transforming growth factor (TGF) beta signaling pathway.

**Figure 7 toxins-15-00594-f007:**
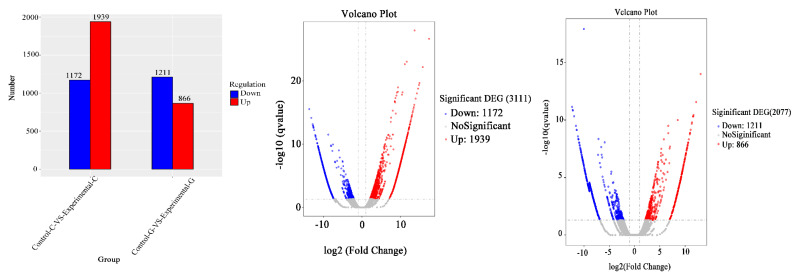
Differential comparison of gene expression up-regulation and down−regulation in samples and differential gene volcano map.

**Figure 8 toxins-15-00594-f008:**
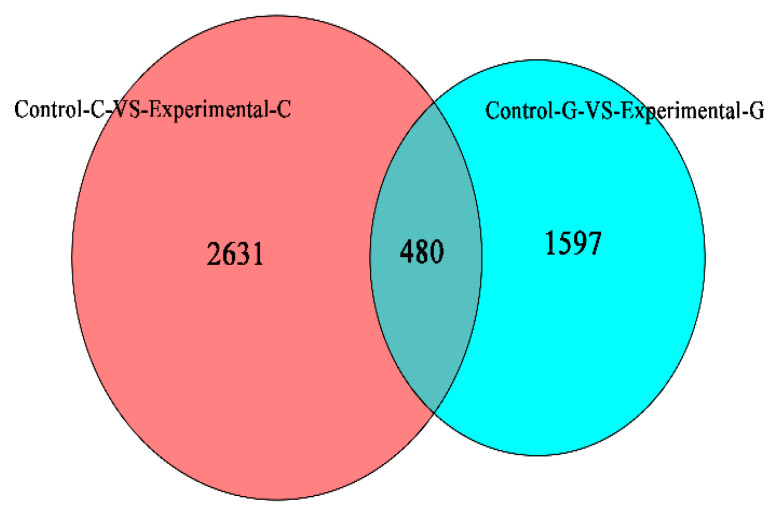
Differential gene Venn diagram.

**Figure 9 toxins-15-00594-f009:**
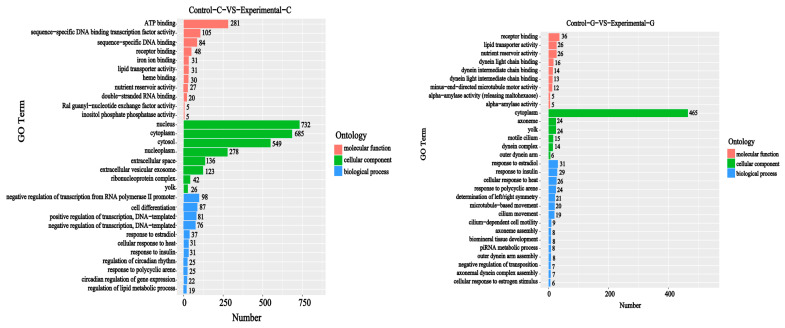
GO enrichment histogram of sample differential comparison genes. The vertical axis represents the enriched GO term, and the horizontal axis represents the number of differential genes in the term. Different colors are used to distinguish biological processes, cellular components, and molecular functions.

**Figure 10 toxins-15-00594-f010:**
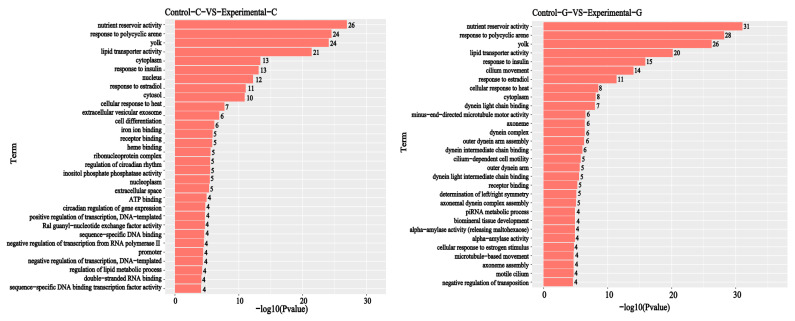
Sample difference comparison gene GO enrichment *p* value histogram, with the vertical axis representing the enriched GO term and the horizontal axis representing the log10 (*p*−value) value.

**Figure 11 toxins-15-00594-f011:**
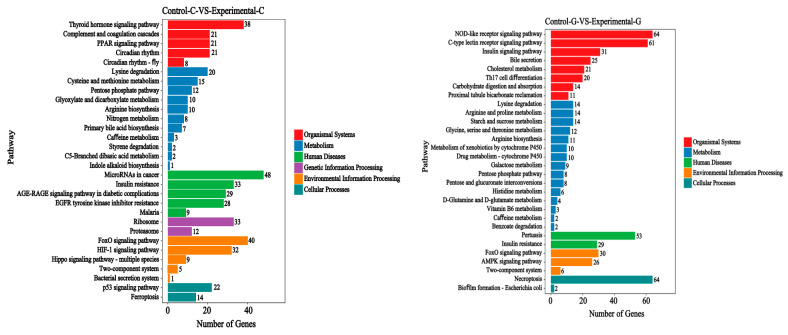
Bar chart of KEGG annotation classification with significantly enriched genes for sample difference comparison; the vertical axis represents the name of the path, and the horizontal axis represents the number of genes. The advanced glycosylation end products—receptors (AGE−RAGE) signaling pathway in diabetic complications, forkhead box O (FoxO) signaling pathway.

**Figure 12 toxins-15-00594-f012:**
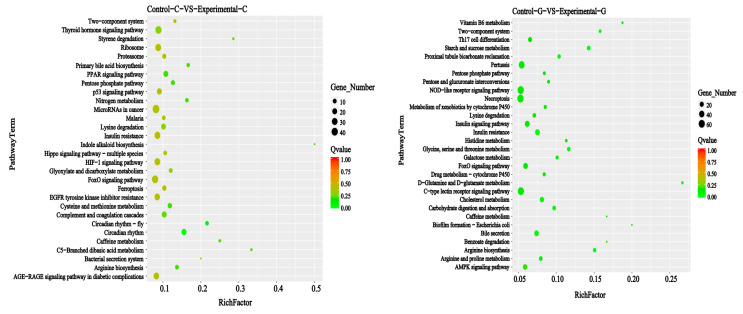
Scatter plot of KEGG enrichment in the comparison of sample differences. The vertical axis represents the path name, the horizontal axis represents the Rich factor, and the size of the dots represents the number of differentially expressed genes in this path. The colors of the dots correspond to different Q value ranges. The advanced glycosylation end—product advanced glycosylation end (AGE−RAGE) signaling pathway in diabetic complications, adenosine 5‘-monophosphate activated protein kinase (AMPK) signaling pathway.

## Data Availability

The data presented in this study are available on request from the corresponding author.

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
