# Peer review of "Transcriptome Analysis of Deoxynivalenol (DON)-Induced Hepatic and Intestinal Toxicity in Zebrafish: Insights into Gene Expression and Potential Detoxification Pathways"

_toxins, 2023, doi:10.3390/toxins15100594_

Round 1
Reviewer 1 Report
The authors has analyzed the transcriptome of deoxynivalenol-Induced hepatic and intestinal toxicity in zebrafish. The topic is interesting. The experimental was well-desgsined. The findings will help to understand the toxicity of DON, which could guide future research in biological detoxification strategies. The following changes could improve the quality of the paper.
1. Lines 8-10, such as GO and KEGG etc., the abbrevations need to be writen with the full name when they are first appereed in the paper.
2. Lines 44-46, please add the recently references about the occrences of DON in the feeds of China. Such as: Occurrence of Aflatoxin B1, deoxynivalenol and zearalenone in feeds in China during 2018–2020.Journal of Animal Science and Biotechnology. 2021.
3. Lines 55-68, please add the recently references about the novel toxicity of DON. Such as: Ferroptosis is involved in deoxynivalenol-induced intestinal damage in pigs. Journal of Animal Science and Biotechnology. 2023.
4. Figures, please increase the quality of the presentation of the figures.
5. there are 12 Figures, please combinded some of them, which will makes the paper seem more compactive.
6. Please add the replicates n=?, and some of the necessary explantion for the professional abbrevations etc. in the figure lengeds.
7. Line 362--, please provid the rational of the DON dose used for the trial by citing the papers.
8. Did you recored the feed intake, weight gain, and other biochemistry data for the Zebrafish?
9. Please check the gramma of the paper.
10. Please check the references and make sure they are followed the followed the standard of journal style.
Author Response
Dear reviewer,
Thank you for your review and suggestions. Please see the attachment point to point answers.

Reviewer 2 Report
The manuscript describes the changes in lncRNA and mRNA expression on liver and intestine of zebrafish after DON exposure. It is an interesting Project but the manuscript is not ready to be published.
Authors must follow the “Author guidelines” of the journal:
https://www.mdpi.com/journal/toxins/instructions
The abstract is long and too general. There are 395 words when they should be 200 maximum. The duration of the exposure should be indicated. Moreover, little information is provided about the most meaningful pathways and DEGs found in the bioinformatic analysis in results and conclusions. In transcriptomics, number of DEGs do not provide the most important information, but the especific pathways and genes related to the organs analysed. A careful interpretation is always needed.
Material and methods section is incomplete, it only explains the exposure of the zebrafish.
In transcriptomics, researchers must always confirm by PCR the expressions changes of some genes of their choice. Here authors can read an example:
Zhang, C., Li, C., Liu, K., & Zhang, Y. (2022). Characterization of zearalenone-induced hepatotoxicity and its mechanisms by transcriptomics in zebrafish model. Chemosphere, 309, 136637.
https://doi.org/10.1016/j.chemosphere.2022.136637
I encourage authors to resubmit when the PCR experiments are performed and the manuscript is revised.
Author Response

(The authors gave the same response as above.)

Reviewer 3 Report
Introduction:
the research question or objective is clearly stated, but relevant sources and studies are missing and references are out of date.
Material and methods:
Fish care and experimental design should be clearer.
miss all IncRNA and mRNA protocols
approval from the institution is required to work with animals.
Results:
improved logical flow and clear section headings, it's hard to see what the most important takeaways are from all the information.
the reading results should be organized by connecting all the figures of "Go to Enrichment Histogram" and them discuss these results. Do the same to "Scatter Plot", "Differential Gene Venn Diagram" figures etc.
the way the figures are presented makes it difficult to understand the most important results.
Discussion:
The section between lines 259 and 271 should be included part in the introduction.
between lines 272 and 277 are the results
line 308 misses units
Author Response

(The authors gave the same response as above.)

Round 2
Reviewer 2 Report
The authors have improved the manuscript following most of the recommendations. I still have two suggestions that would be important to take care of.
1. In the abstract, there are still some sentences that could be changed for the specific results obtained in this work. In my opinion, authors should rewrite this sentences:
“The mRNA of differentially expressed genes (DEGs) based the Gene Ontology (GO) functional annotation analysis, the top 30 KEGG signaling pathways based on the Kyoto Encyclopedia of Genes and Genomes (KEGG) signaling pathways of the top 30 genes were analyzed, respectively.” I propose: Differentially expressed genes (DEGs) from mRNA and lncRNA were analysed by RNA seq. There were studied Gene Ontology (GO) and signaling pathways where the top 30 DEGs of each type of RNA were involved.
“GO enrichment analysis found 370 items in the gut and 160 items in the liver including biological processes, cell composition, and molecular function.” BP, CC and MF are in fact the three main categories of gene ontology, please change that for specific GOs.
“KEGG analysis revealed 2834 signal pathways including biological systems, metabolism, human diseases, environmental information processing, and cell processes in the gut samples.” Instead of KEGG, write Signaling pathways and specify more the name of the main pathways found.
2. Regarding Material and methods, I still think there is information missing. The sequencing was performed by Genewiz enterprise, using Illumina technology as they describe in this technical specifications sheet.
https://fs.hubspotusercontent00.net/hubfs/3478602/13002-SD-2%201221%20RNA-Seq%20Technical%20Specs%20Sell%20Sheet.pdf
It needs to be written in the manuscript that the platform was Illumina Hiseq. The authors indicate GWZHISEQ01 but this is just a code given by the company, not the commercial name of the platform.
Author Response

(The authors gave the same response as above.)

Reviewer 3 Report
This new review of the article answered all my questions.
Author Response
Dear reviewer,
Thank you very much for your review of the manuscript.
the authors